# A New Method of Disabling Face Detection by Drawing Lines between Eyes and Mouth

**Chongyang Zhang** [1,*] and **Hiroyuki Kameda** [2]

1  The Graduate School of Bionics, Computer and Media Sciences, Tokyo University of Technology, 1404-1 Katakuramachi, Hachioji City 192-0982, Tokyo, Japan
2  The School of Computer Science, Tokyo University of Technology, 1404-1 Katakuramachi, Hachioji City 192-0982, Tokyo, Japan
*  Correspondence: zcylno@gmail.com

**Abstract:** Face swapping technology is approaching maturity, and it is difficult to distinguish between real images and fake images. In order to prevent malicious face swapping and ensure the privacy and security of personal photos, we propose a new way to disable the face detector in the face detection stage, which is to add a black line structure to the face part. Using neural network visualization, we found that the black line structure can interrupt the continuity of facial features extracted by the face detector, thus making the three face detectors MTCNN, S3FD, and SSD fail simultaneously. By widening the width of the black line, MTCNN, S3FD, and SSD are able to reach probability of failure levels up to 95.7%. To reduce the amount of perturbation added and determine the effective range of perturbation addition, we firstly experimentally prove that adding perturbation to the background cannot interfere with the detector's detection of faces.

**Keywords:** filtering method; image discrimination and classification; human image processing; face detection; adversarial attack





## 1. Introduction

Disinformation has flooded social media recently, affecting people's ability to judge some events correctly. False information mainly includes fake text information and fake videos. For example, using deepfake technology to change the faces of politicians and make inappropriate speeches is enough to cause a crisis. Although existing research has been able to identify fake videos well [1], when users browse short videos on social software, they will immediately choose to believe them instead of using other programs to identify the authenticity. Therefore, it is imperative to prevent the occurrence of face swapping fundamentally.

The currently popular face-swapping program, *faceswap*, can replace a face in a video through three steps: face detection and positioning, training of the deepfake [2] model, and face fusion. By analyzing the face-swapping process of *faceswap*, we propose that if the face detector fails to detect a face in the image, the face-swapping behavior can be prevented.

*faceswap* provides three optional face detectors: the Multi-Task Cascaded Convolutional Neural Network (MTCNN) [3], the Single Shot Scale-invariant Face Detector (S3FD) [4], and the Single Shot Multi-Box Detector (SSD) [5] based the OpenCV deep neural network module [6] in face detection and positioning. After face detection on the image, the user can manually delete some misrecognized faces. From this, we believe that the failure of the face detector should be the generation of no bounding box at all, rather than merely shrinking or shifting the bounding box.

An adversarial attack involves adding some imperceptible, subtle perturbations to the input data to make the model give a wrong output with high confidence [7]. The attack target can be a classification model or a target detection model. The attack range is generally the entire image when attacking a classification model. Although no research has pointed

out that only the attack target is effective, when attacking the target detection model, the area where the target is located is generally selected for attack. In the research on attacking face detectors, we found that some studies reduce the detection probability of faces by increasing the detection probability of non-face areas [8]. Therefore, before experimenting, it is necessary to find the effective attack range in the image.

Adversarial attacks can quickly attack SSD and S3FD based on Region Convolutional Neural Networks (RCNN) [9]. Even without adjusting the parameters, the added perturbation is difficult to detect by humans [10]. However, since MTCNN uses the image pyramid [11] to process the input image, its defense capability far exceeds the above two face detectors. The image pyramid is a method of reducing an image and can reduce the original image to multiple sizes. It prevents the perturbation at the original size from interfering with the detection results of images at other sizes. The most effective way to attack MTCNN currently in the field of physical attacks is for the researcher to wear a black hat and research, or wear a mask, and then put two black and white patches on the face. However, such images cannot be exploited and uploaded to social media. In the digital field, researchers interpolate the perturbations at different sizes according to the characteristics of the image pyramid, and finally obtain effective perturbations [12]. However, this method cannot attack three face detectors simultaneously.

We analyze and verify the effective range of attacking face detectors, propose a black line structure that can simultaneously invalidate MTCNN, S3FD, and SSD face detectors, and, through neural network visualization, the reasons that the black line structure is effective are analyzed.

This article's remainder is structured as follows: Section 2 introduces the related works and analyzes and compares its principles. Section 3 describes our experiments, which are the face–background correlation analysis experiment, the black line structure experiment, and the black line structure validity analysis and discussion. Section 4 will summarize and refer to future work.

## 2. Related Works

### 2.1. Face Detection

#### 2.1.1. Multi-Task Convolutional Neural Network

MTCNN consists of three networks: Proposal Network (P-Net), Refine Network (R-Net), and Output Network (O-Net). P-Net mainly obtains the candidate bounding box of the face region, calibrates the candidate bounding box, merges highly overlapping candidate bounding boxes through non-maximum suppression (NMS) [13], and passes the result as input data to R-Net. R-Net removes the false-positive regions through bounding box regression and the NMS, but the network structure is slightly different from P-Net, with an additional fully connected layer. O-Net has one more volume-based network than R-Net, so the processing results will be more refined, and the data used are the output data of R-Net. O-Net will eventually output a face bounding box and five facial keypoints, including eyes, mouth corners, and nose.

P-Net is the fully convolutional network structure (FCN) [14], which can simultaneously process images of various sizes. The data will be processed using image pyramids before being passed to P-Net. The image pyramid is controlled by two parameters, the minimum size and the scaling factor. The minimum size represents the side length of the minimum image, and the scaling factor is the ratio of scaling the image each time.

#### 2.1.2. Single Shot Multi-Box Detector

SSD uses the feature pyramid to enhance the detection ability of the model, and all six networks inside the model can contribute to the final detection result. Moreover, SSD uses the FCN to extract the information transmitted by the feature pyramid, which improves the calculation speed.

### 2.1.3. Single Shot Scale-Invariant Face Detector

Due to the weak ability of SSD to detect small faces, the researchers based on SSD proposed a small face detection algorithm, S3FD. They set the minimum stride to 4 and introduced a scale-compensated anchor matching strategy to increase the number of positive anchors to improve the detection ability of small faces.

### 2.1.4. Convolutional Neural Networks

Lecun [15] borrowed ideas from Fukushima [16] and proposed the original version of the convolutional neural network, LeNet [17], in 1998. The essence of the convolutional neural network is a feature extraction method. The sliding filter can share the weights while ensuring the local receptive field's information integrity, and the pooling layer can effectively reduce the amount of calculation.

In face detection work, MTCNN, SSD, and S3FD have different characteristics, but they all use the convolutional neural network for image feature extraction. This method of feature extraction using a sliding window can continuously and completely extract image information.

### 2.2. Interpretability of Neural Networks

The neural network works in an end-to-end manner, and we can achieve the desired results without understanding its complex internal work. However, to be able to use neural networks better, researchers began to explore methods that can explain the internals of neural networks [18–21]. These methods can help us to understand which pixels in the image affect the model's decision making. The iNNvestigate project [22] integrates 13 types of neural network visualization methods, which are used in the present work to accurately visualize our results and to conduct comparative analysis.

### 2.3. Neural Style Transfer

Neural style transfer involves splitting an image into content and style and combining the content with the style of another image to obtain a new image. When a multi-layer convolutional neural network performs feature extraction on an image, the features extracted by each network layer are different [23]. The high-level network extracts content information, and the low-level layer extracts texture information—that is, style information.

## 3. Experiment and Discussion

### 3.1. Correlation of Background and Face (Experiment 1)

Li et al. [8] attacked the face dataset by adding perturbations to the entire image to make the face detector fail and detect non-human face parts so that the face replacement cannot obtain valid data. However, *faceswap* does not need to use the dataset, but two selected videos. Moreover, we believe that adding perturbation to the whole image may cause unnecessary pollution to the image. We will demonstrate through this experiment whether it is necessary to add perturbations to the background when attacking the face detector.

### 3.1.1. Method and Results

To make the comparison of experimental results explicit, the threshold of P-Net in MTCNN is set to 0.9 to reduce the candidate boxes. As shown in Figure 1, we use P-Net to detect the image roughly and obtain bounding boxes of the face and background parts. Afterward, neural style transfer combines facial features with background content to enhance the number of facial features in the background bounding box, as shown in Figure 2. We use P-Net for detection, and Figure 3 shows the final result.

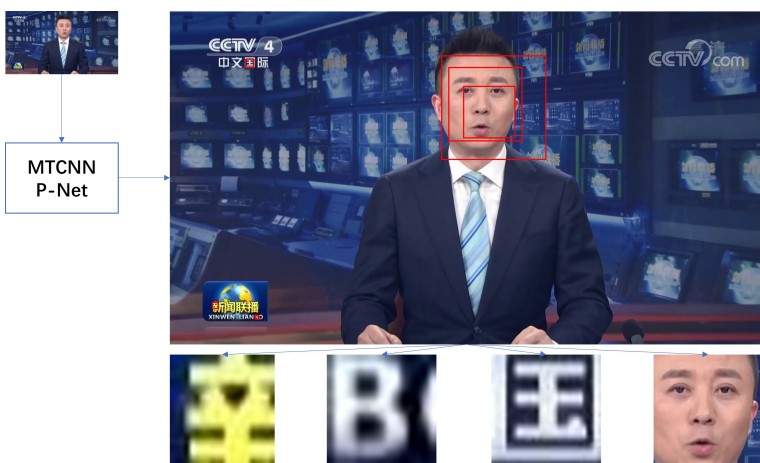

**Figure 1.** Use of MTCNN's P-Net network for rough detection and extraction of the bounding box content. Three non-face regions are extracted from the detected bounding boxes, as well as an image of a face region. The non-face regions contain characters that are caused by the wrong detection of the detector. This experiment only considers him to be a part of the background and has nothing to do with its actual meaning.

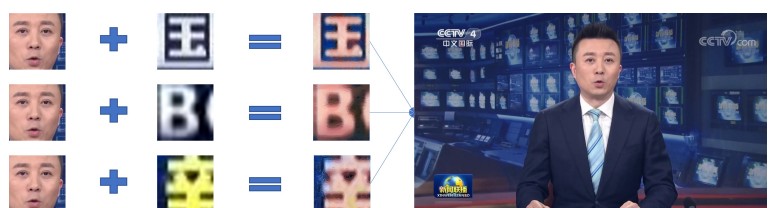

**Figure 2.** The images of the face region and the images of the three non-face regions are style transferred and restored to their original positions.

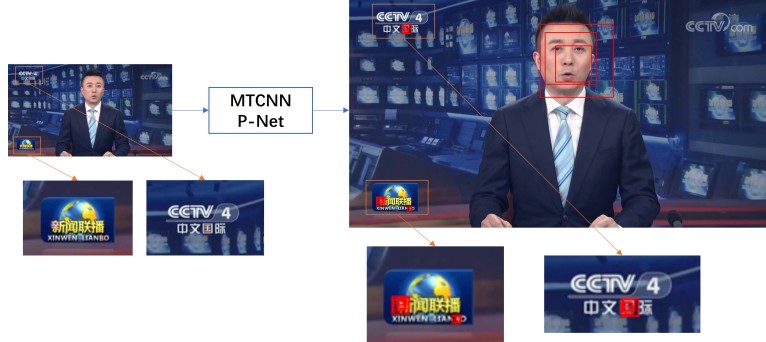

**Figure 3.** After the transfer of the local style, the original image and the image are used for face detection using MTCNN. Although the detection probability of the part of the style transfer increases, the number of face detection bounding box does not decrease.

### 3.1.2. Discussion

The results show that the detected bounding boxes increase significantly in the background region fused with facial features, but there is no change in the facial area. Thus, when attacking the face detector, only the face in the image needs to be attacked, and it is meaningless to add perturbation to the background. It is not difficult to understand that when detecting a multi-face image, the feature information provided by each face in the image is independent.

### 3.2. Adding Black Line Structure (Experiment 2)

As shown in Figure 4, the use of black patches to block facial keypoints such as eyes can still be detected by the three face detectors. However, masking the areas between facial keypoints with black line segments can disable the face detector. We found that MTCNN, SSD, and S3FD are based on CNN and use a slider window to extract image features, and the black line segments are likely to interrupt the continuity of features, thus causing the failure of these three face detectors. To demonstrate the effectiveness of the black line structure, we conduct experiments.

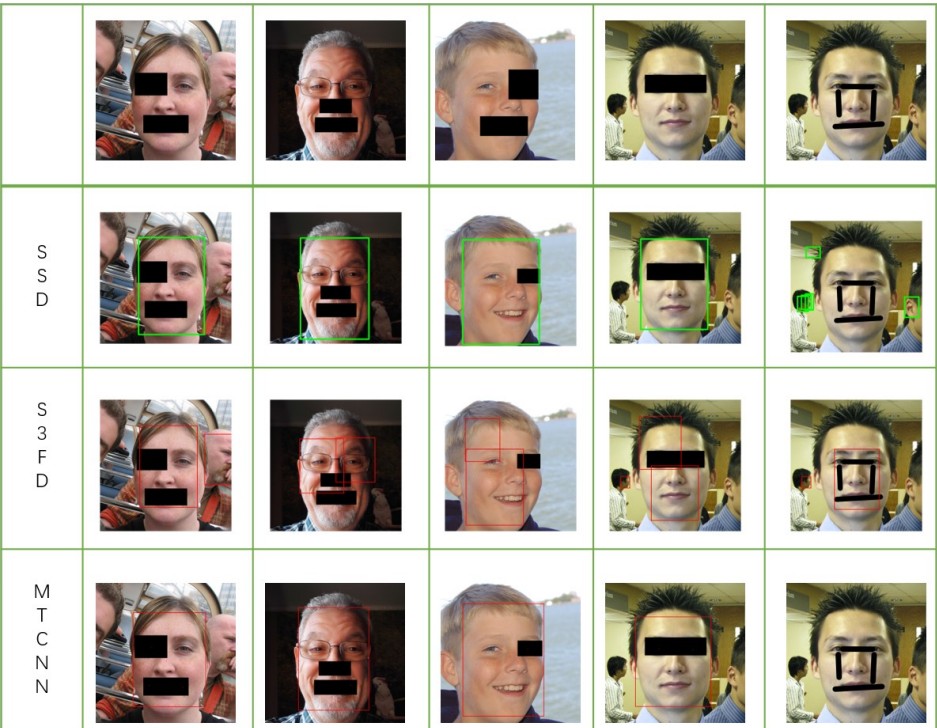

**Figure 4.** Blocking part of the facial area and performing face detection through face detectors SSD, S3FD, and MTCNN.

### 3.2.1. Datasets

We choose the CelebFaces Attribute (CelebA) [24] dataset and Flickr-Faces-High-Quality (FFHQ) [25] dataset. The CelebA dataset is openly provided by the Chinese University of Hong Kong and is widely used for face-related computer vision training tasks. We extracted 10,000 images from 202,599 for this research. The FFHQ dataset was created as a benchmark for GAN. NVIDIA open-sourced it in 2019. It is a high-quality face dataset. We randomly extracted 6000 images from among 70,000 images for this experiment. In this way, we have an actual multi-scene face dataset and a dataset enhanced by GANs to ensure the universality of this experiment.

### 3.2.2. Method

Among the three networks of MTCNN, the threshold is 0.6 for P-net, 0.6 for R-Net, and 0.7 for O-Net. The minimum size of the image pyramid is 21 pixels, with a scaling factor of 0.709. The threshold for SSD and S3FD is 0.6. These parameters are verified in a large number of experiments.

As shown in Figure 5a, a few simple hand-drawn black lines can make the three face detection algorithms fail. There are many such images, but the position and thickness of the black line in each image are different. Thus, we propose a normalization method to add black lines, as shown in Figure 5b.

Since we need to use the locations of facial keypoints, we use MTCNN to find bounding boxes and facial keypoints. We use black line segments to connect the upper left vertex of the bounding box to the left eye, the upper right vertex to the right eye, the lower left vertex to the left corner of the mouth, and the lower right vertex to the right corner of the mouth. Finally, we connect the eyes and the corners of the mouth to form a rectangle. Due to the different sizes of the CelebA and FFHQ data, we added a black line structure with pixel widths of 6 pixels, 8 pixels, and 10 pixels to the CelebA data and a black line structure with a pixel width of 4 pixels, 6 pixels, and 8 pixels to the FFHQ data, respectively.

For the detection results, we must be consistent with the limits of *faceswap*. If no bounding box is detected at all, the attack is considered successful.

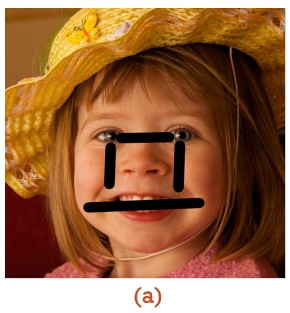 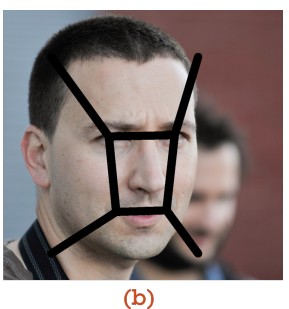

(a)         (b)

**Figure 5.** Image (**a**) is a manually added black line break. Image (**b**) is a presentation image of the black line structure. Both images are from the FFHQ dataset.

### 3.2.3. Result

Tables 1 and 2 and Figure 6 show the results of this experiment. We use whether the face detector returns bounding boxes as a benchmark for detection accuracy. Even if the returned bounding box is inaccurate, the attack is counted as a failure. This is because the *faceswap* program provides a manual calibration step; as long as there is a bounding box returned, it will be provided to this step. Shown in the table and image is the probability that a face in the image can be detected.

**Table 1.** The table shows the detection capabilities of the three face detectors on the original images in the CelebA dataset and on images with black line structures of 6-pixel, 8-pixel, and 10-pixel widths added.

| CelebA | Original | 6 Pixels | 8 Pixels | 10 Pixels |
|---|---|---|---|---|
| S3FD | 99.67% | 65.5% | 49.4% | 32.1% |
| SSD | 99.71% | 24.9% | 15.08% | 9.47% |
| MTCNN | 99.79% | 9.08% | 7.49% | 6.15% |

**Table 2.** The table shows the detection capabilities of the three face detectors on the original images in the FFHQ dataset and on images with black line structures of 4-pixel, 6-pixel, and 8-pixel widths added.

| FFHQ | Original | 4 Pixels | 6 Pixels | 8 Pixels |
|---|---|---|---|---|
| S3FD | 99.7% | 58.7% | 23.7% | 9% |
| SSD | 99.93% | 22.4% | 9.7% | 4.3% |
| MTCNN | 99.96% | 11.7% | 8.8% | 7.2% |

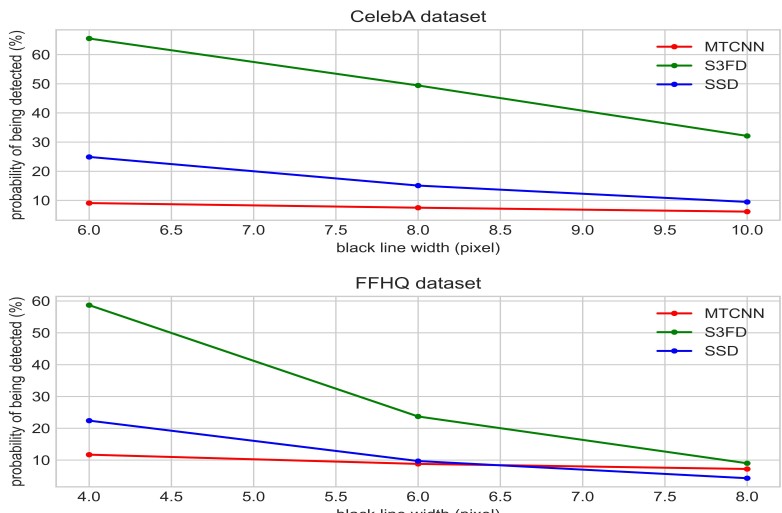

**Figure 6.** Different unit pixel widths are used due to the other face sizes in the CelebA dataset and the FFHQ dataset. The table shows the probability of the black line image being detected by MTCNN, S3FD, and SSD.

### 3.2.4. Discussion

Experiments show that the black line structure can invalidate MTCNN, SSD, and S3FD, with failures ranging from 34.5% to 95.7%. Even if the bounding box can be generated, it is difficult to detect the entire face, as shown in Figure 7.

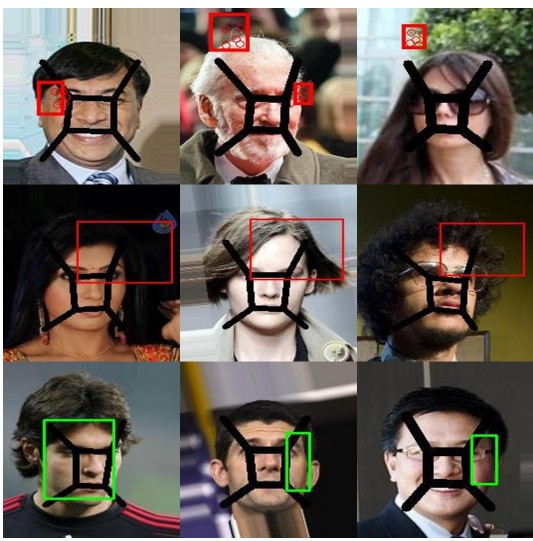

**Figure 7.** The image above shows some images with black line structure added but which can be detected by the face detector.

We used iNNvestigate to perform neural network interpretability analysis on images with a black line structure. In Figures 8–10, the images of successful and unsuccessful attacks under different black line widths are compared. The first to third images in the first column on the left are the images that the face detector cannot detect, and the fourth to sixth images are the images that the face detector can detect. It can be found that the effective black line structure is more obvious in the visualization and can segment the face. The invalid black line structure is relatively blurred in the visualization, and the degree of integration with the face is relatively high. Thus, we can be sure that when the facial features are cut, the features read by the model cannot be associated with the information of the facial keypoints, which will cause the face detector to fail. It also proves that breaking the continuity of features between facial keypoints can invalidate the face detector.

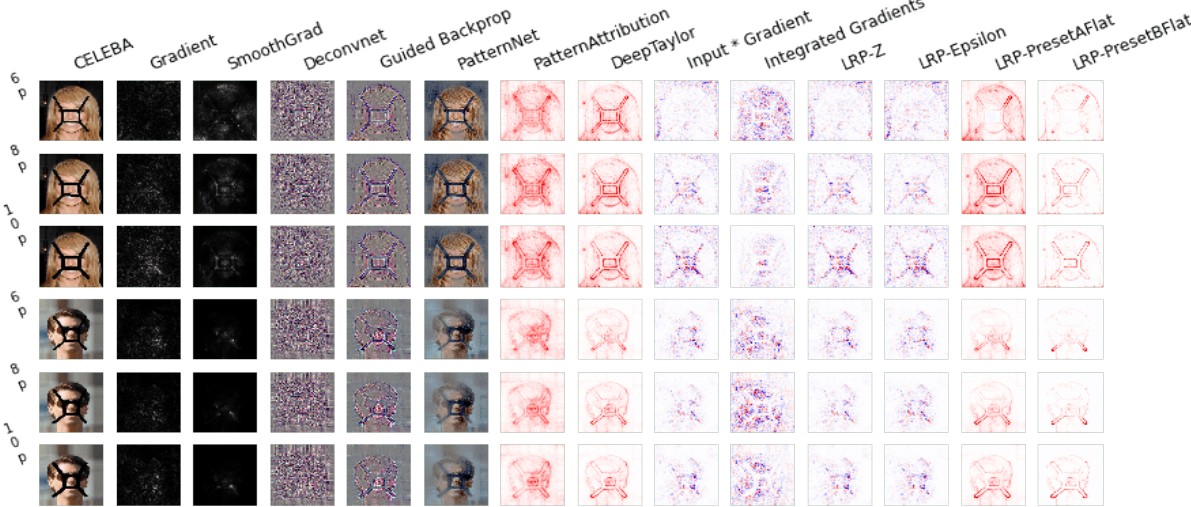

**Figure 8.** We selected two images from the CelebA dataset and added a black line structure with widths of 6 pixels, 8 pixels, and 10 pixels. MTCNN cannot detect the first three pictures, and the last three pictures are pictures that MTCNN can detect.

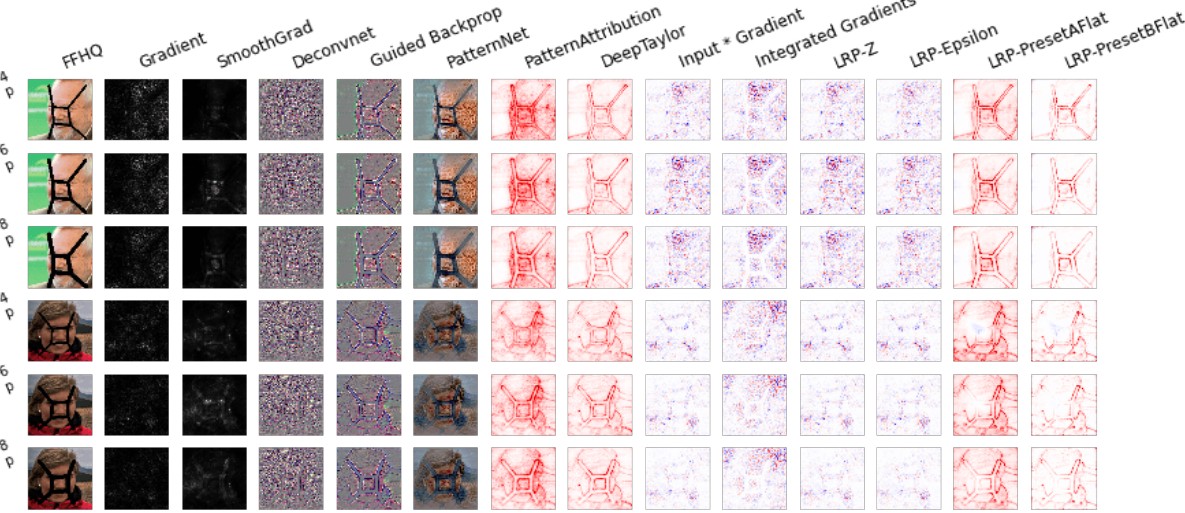

**Figure 9.** We selected two images from the FFHQ dataset and added a black line structure with widths of 4 pixels, 6 pixels, and 8 pixels. S3FD cannot detect the first three pictures, and the last three pictures are pictures that S3FD can detect.

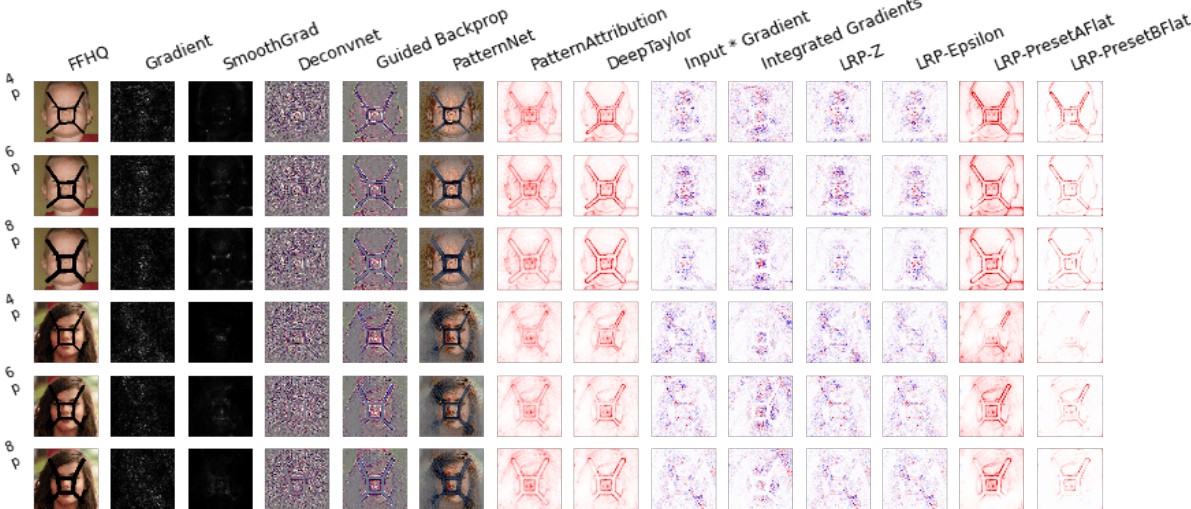

**Figure 10.** We selected two images from the FFHQ dataset and added a black line structure with widths of 4 pixels, 6 pixels, and 8 pixels. SSD cannot detect the first three pictures, and the last three pictures are pictures that SSD can detect.

## 4. Conclusions

We demonstrate that attacking the background is ineffective when attacking face detectors and show the black line structures that may invalidate MTCNN, S3FD, and SSD. However, it must be noted that although the black line structure can disable the face detector, it does not meet users' needs regarding uploads to social media because it does block the face. Just as hand-painted black lines can achieve the same effect, there must be excess disturbance in the parts covered by the black line structure. This research aims to encourage more researchers to pay attention to the nature of adversarial attacks through the success of the black line structure. At present, most of the mainstream face detectors are based on CNNs. If the continuity of CNN extraction features is interrupted, it will be possible to find a general method to attack face detectors. Next, we will add perturbation to the black line area to find a balance that prevents face swapping and permits the use of the image.

**Author Contributions:** Writing—original draft, C.Z.; Writing—review & editing, H.K. All authors have read and agreed to the published version of the manuscript.

**Funding:** This research received no external funding.

**Data Availability Statement:** All data were presented in main text.

**Conflicts of Interest:** The authors declare no conflict of interest.

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
