# Peer review of "A New Method of Disabling Face Detection by Drawing Lines between Eyes and Mouth"

_computers, doi:10.3390/computers11090134_

Round 1
Reviewer 1 Report
The paper attempts to develop an approach to defeat attempts to create Deep Fake images. Deep Fakes are created by identifying faces and then replacing faces in other images, or creating composite faces.
The approach involves superimposing a grid of black lines over the face. The authors have experimented with lines of differnet widths. They have used selected 600 images from the CelebA and FFHQ data bases. The modified images are examined using Multi-task Convolutional Neural Network (MTCNN), Single Shot MultiBox Detector (SSD) and small face detection algorithm (S3FD) to identify facial features. They find that width of the black line is large, the probability of feature detection can be reduced significantlly. For example, with a line 8 pixels wide, the probability of detection drops to 4.3 to 9%.
In realted work such as [7], adversarial perturbations have been used to defect feature identification while the perturbations in the image remains imperceptible. However in this scheme, the black lines are clearly seen, defeating the objective of the sharing the image, whatever it might be. The authors acknowledge this disadvantage.
The authors suggest that "we’ll add perturbation to the black line area to find a balance that prevents face swapping and uses the image." I don't see how that would be done.
It is not at all clear that this work will result to any practical use, or will lead to further work that will yield useful results. The looks like a brute-force method to defeat facial recognition.
There are some practical situations where obscuring selected faces, often by fuzzing, is done in posted images. However it looks to me that this approach will not defeat face identification for a human observer who is familiar with the faces.
Author Response
Thank you for your suggestion.
As you said, we admit that the black line structure is recognizable by the naked eye. Therefore, in order to make the added perturbation invisible to the human and to be different from the previous attack methods, we hope to add perturbation within the range covered by the black line, rather than violently covering it with pure black. Such as replacing black with randomly generated perturbations. It is also our next work content.
Another important point is why we propose attacks that break the continuum of CNN feature extraction. I added this part to the introduction and cited some new papers. MTCNN is a special face detector, and it is difficult to attack successfully in the digital domain. The black line structure differs from the general adversarial attack. The black line structure can attack multiple face detectors simultaneously.
Reviewer 2 Report
The paper presents a new way to disable the face detector in the face detection stage, which is to add a black line structure to the face part. Using neural network visualization, we found that the black line structure can interrupt the continuity of facial features extracted by the face detector, thus making the three face detectors MTCNN, S3FD, and SSD fail simultaneously. By widening the width of the black line, the probability of failure of MTCNN, S3FD, and SSD is up to 95.7%
This paper presents work in an important field and to the best of my knowledge the paper is original and unpublished.
In the present form I suggest to submit the paper to a conference to get advices and suggestions on how to develop the experiments in order to recognize deep fake videos.
Author Response
Thank you for your suggestion
In order to improve the overall structure and smoothness of the paper, we have revised the paper. To demonstrate the importance of the research and the novelty of the idea, we have added some content to the introduction, experiments, discussions, etc.
We are looking forward to your new suggestions.
Reviewer 3 Report
The author introduces a new filter that can work against deep fake in this paper nicely; the interest in the topic is current but still needs much improvement to move further: 1) The abstract is not clear. Also, please expand the abbreviation in the abstract and anywhere in the manuscript if it appears for the first time. 2) The title talks about the filter; it is hard to understand what kind of filter or the name of the filter the author is claiming that they proposed. Introduction: 1) The introduction is not well written. I need to make it more composed. 2) Again, please elaborate on the abbreviation in lines 26 and 27. This is very frustrating.. the author mentioned dnn.. what is the author expecting? dnn should be a deep neural network. 3) Introduction is too small, and not enough information is provided. Related work: 1) The entire related work needs to revise. There is not a single previous manuscript about deep fake. I would suggest the author should read some of the previous literature carefully in order to understand what is the literature review. The author might add the following papers in the review section: 1) Gupta, K. D., Ahsan, M., Andrei, S., & Alam, K. M. R. (2017). A robust approach to facial orientation recognition from facial features. BRAIN. Broad Research in Artificial Intelligence and Neuroscience, 8(3), 5-12. 2) Lyu, S. (2020, July). Deepfake detection: Current challenges and next steps. In 2020 IEEE international conference on multimedia & expo workshops (ICMEW) (pp. 1-6). IEEE. Experimental and discussion: 1) This isn't very clear. The author later added methods and results. Please have a look at other papers. 2) There is no indication of what kind of statistical measurement has been used. 3) No information on the dataset. 4) I doubt this work. For fair act reproducibility, the author should make their code publicly available. There is a severe flaw in their experimental result presentation and what they are claiming. 5) I do not see where the author discusses any topic Conclusion: 1) After reading the entire paper, I did not find whether the new filter was introduced. 2) Please revise the conclusion.Author Response
Abstract
1), 2)
Thanks for the suggestion. I expanded the abbreviation in the introduction.
The filter in the title does not refer to the neural network filter but the black line structure.
The image with the black line structure added can be deepfake filtered or ignored by the face detector so that the face is not changed.
Introduction
1). 2). 3)
I have added something to the introduction to make our research even more important.
The dnn here is not what you mean by the deep neural network, but the dnn module in the OpenCV package. Sorry, I didn't comment on it. I've improved it.
Related work
1)
Thank you for your suggestion.
It may be that some content in the introduction is not clearly stated, causing your misunderstanding.
Although this paper aims to invalidate the deepfakes, it is achieved by invalidating face detectors.
Therefore, we only added references directly related to this experiment in the prior study.
In related work, I added some details.
If the content of the changed introduction still does not clear your doubts, we can consider adding a section about the deepfakes.
Experimental and discussion
1)
I made some modifications to the expression part of the experimental results.
2)
Experiment 2 uses whether to return the bounding box as a statistical method. If no bounding box is returned at all, the black line structure on this image is judged to be valid. Even if the deviation is large, as long as a bounding box is returned, it will be judged that the black line structure on this image is invalid, as shown in Figure 7. This statistical method, introduced in the introduction, is also specified in 3.2.2.
3)
This paper uses two datasets, CelebA and FFHQ, which are introduced in 3.2.1.
4)
The code can be made public, but we don't think it's necessary. Because the code is simple, it is described in detail in the method. The face detection algorithms we use are the most popular projects on GitHub, and the parameters are consistent. The difference is the python code for the line drawing part. For ease of reproduction, we added threshold parameters for each model in 3.2.2.
5)
In the discussion of the two experiments, we discuss the face as the only important region in aggressive behavior and the effectiveness of the black line structure in attacking three face detectors, respectively. In the first discussion, we have identified the target of the attack, so there will be experiment 2 that only constructs the black line structure on the face. And through Experiment 2, we found a different way to filter images than generative adversarial attacks, which can disable face detectors and thus deepfakes.
Conclusion
1), 2)
The black line structure is the new filtering method so that the image passing through the face detector is filtered out.
However, due to the mismatch between the original title and the title, we decided to change to a more suitable title for the experiment.
Thanks for your advice.
Reviewer 4 Report
In short, I really enjoyed the authors' idea of putting continuous black lines inside face to prevent face detection. But the way of organizing the authors' brilliant idea is so bad. English usage should be severely improved; extensive details on the background knowledges should be added; and experimental results should be explained more carefully with better organization. Also, I'm not sure whether adding black line can be treated as 'filtering' and I don't know why 'Deep Fake' is in the title given that the authors did not show any experimental results regarding 'Deep Fake'. Lastly, and most importantly, precise description on how to draw the 'black lines' are missing. Please find my details comments below.
1. Missing 'Proposed Solution' section.
After Sec. 2 'related works', Sec. 3 'experiment and discussion' came out from nowhere. This is too digested way of writing a paper. Please add explicit section that describes proposed solution itself. Also, the precise way of drawing black lines is missing. One can just draw connected line that passes eyes and mouth? Fig 5 shows that it seems to also draw until the line meets the boundary of the face. Please give precise method on how to draw this black line. Maybe using some graph-theoretical description would be beneficial, but at least there should be detailed guideline on how to draw them.
2. Not clear what Fig. 1~3 try to show.
It seems that the authors tried to show that what really matters is the facial part not the background. But isn't this so trivial thing? Is there any works that prevent deep fake by doing something on the background? In current way, it's not clear why the authors added Fig. 1~3.
3. Experimental proof on 'no affects in the background' is not clear from Fig. 1~3.
The authors claim that perturbing the bounding boxes in the background does not affect face swapping. But the 'experimental proof' is not clear. What is the meaning of style transfer of the bounding boxes of the background? And what does it infer that the 'detection probability' has been increased? How do you define 'detection probability'? Is it the probability of making the bounding box correctly or the actual classification based on the bounding box? If the authors want to prevent deep fake, then shouldn't the perturbation be considered also outside the bounding box? Since bounding box is performed based on the pure image, adding perturbation to the bounding box seems not straightforward. Please clarify why the authors particularly chosen such setting.
4. Results relating to deep fake, or face swapping is missing.
Currently, the authors are just focusing on face detection. The results are promising but there is no further investigation on deep fake or face swapping. Without such experiments, this paper is just about face detection. The reviewer thinks that face detection itself is an interesting topic. Please choose either (i) change the whole paper to clarify that this paper is focusing on face detection; (ii) add additional experiments on actual face swapping and how the proposed method can prevent face swapping. I guess the faceswapping result would yield very weird image which verifies that the authors' black line strategy is indeed powerful.
5. In the title, 'filtering' is not the best word I guess. It feels like extracting some useful information from the input which the authors' black line is not doing. Maybe 'protection' instead of 'filtering'? I guess authors would know better expression than me but at least I think filtering is quite misleading.
6. In the title, the authors mentioned 'movies'. But the 'movies' have never been addressed. Remove 'movies' or add experiments regarding movies.
7. In the title, Deep Fake has been mentioned but never explicitly considered. Change to 'face detection' or add experiments on deep fake.
8. Neural network visualization, Fig. 8, has so lack of explanation. I cannot get any information from the current draft. Please detail the figure and express how this is related to the proposed solution.
9. In the abstract, the authors mentioned 'experimentally proved for the first time that adding perturbation to the background cannot interfere with the detector ’s detection of faces.' But this is too strong statement given that the authors just did a single visualization experiment. Weaken this strong statement or provide strong experimental results, e.g., quantitative results, to argue that the authors 'firstly' show this impact.
10) Threshold for MTCNN, SSD are not defined. Please detail them. P-Net, R-Net, O-Net are not explained. Image pyramid is not defined. 'minimum scale is not less than 21 pixels wide' is not understandable. There are so many details are missing like this throughout the paper which prevent in-depth reading. Also, what is Fig. 5 doing?
11) Results in 3.2.3 is not readable. Please summarize in a table.
12) English usage must be improved.
a) 'it is difficult to distinguish between real and fake' -> 'it is difficult to distinguish between real images and fake images' or -> 'it is difficult to distinguish whether the images are real or fake'. Also this sentence itself is not clear.
b) 'By widening the width of the black line, the probability of failure of MTCNN, S3FD, and SSD is up to 95.7%.' -> 'By widening the width of the black line, MTCNN, S3FD, and SSD are able to reach probability of failure level up to 95.7%.'
c) Too many 'us', 'we'. E.g., 'have aroused our concern.', 'hods, which are used for us to a'
d) 'The currently popular face-swapping program faceswap can replace the face in the video through three steps,' -> 'The currently popular face-swapping program, faceswap, can replace the face in the video through three steps,'
e) 'ce fusion, it makes face-swapping easy. ' Not readable.
f) 'dnn' never defined.
g) MTCNN[3], S3FD[4], and SSD[5] -- full name missing even though they are in the related works section but it's hard to connect them.
h) 'And, after face detection on he image, a manual face screening func' -> what is face screening?
i) 2.1.4, defining CNN given that CNN has been addressed in 2.1.1, is not clear.
Again, I liked the authors' idea and believe it can become a good paper.
Author Response
0
Thank you very much for your suggestion.
The scope covered by the previous title is a bit large, ignoring the specific tasks accomplished in this article.
Therefore, we have changed the title, hoping to be closer to the content of this paper.
1
I've added a few things to the introduction, which I hope will serve as a link to the third chapter.
Your observation is correct, our method does cover parts like the eyes and the corners of the mouth.
In fact, each photo has a different black line structure that prevents it from being detected by the face detector.
But in order to achieve a unified way, we used the form described in the text to draw a unified picture.
I've also added a description to the text and an example of a black-line structure that doesn't use the generic method, which is also resistant to detection by the three face detectors.
2
When ordinary researchers conduct adversarial attacks on target detection models, they will focus on the target itself for the first time, and I also think so.
However, the method used in Reference 8 forces us to judge the correct attack range.
Reference 8 makes deepfake invalid by adding perturbation to the whole image so that the face detector can only detect non-face parts.
So, I did a simple experiment to prove that when attacking the face detector, only attacking the face is enough, and improving the detection probability of other parts is invalid.
It allows us to focus our experiments on the face.
3
In this experiment, we use neural style transfer, which aims to transfer the features of faces to non-face regions to increase the detection probability of non-faces.
When the non-face area's detection probability increases, the face's detection probability does not change, which means that the feature change of the non-face area does not affect face detection.
The intuitive manifestation of detection probability is the change in the number of bounding boxes.
4
You're quite right, and we overlook what the experiment achieves.
So we decided to revise the paper's title based on your suggestion, linking the title to the content.
Regarding the result of (ii), I think it is impossible to change faces through the faceswap program.
The face detector provided by faceswap cannot detect faces in images or videos.
Of course, this is also our ultimate goal.
5,6,7
Yes, you are right. I fixed the title.
8
We can clearly see what different networks focus on the image through neural network visualization.
For comparison, I have added a few more neural network visualizations of different images to the text.
It can be found that when the network reads the image, each part of the face is segmented by the black line structure.
We believe that this segmentation can prove that the black line structure blocks the continuity of the network for feature extraction.
Of course, one would think this is bound to happen, but as shown in Figure 4, masking multiple face key locations doesn't stop the face detector either.
However, the black line structure can, so the occlusion method of the black line structure must interfere with the continuity of feature extraction.
9
Thanks for the reminder. I'll pay attention to the wording, and it's been revised.
10
Image (b) in Figure 5 shows a standard black line structure. To illustrate to the reader that the black-line structure has a lot of room for optimization, we add a hand-drawn black-line image (a), whose face is also not detected by the face detector.
I completed the parameters of MTCNN, SSD, and S3FD.
The relevant introduction of the three networks of MTCNN and the relevant introduction of the image pyramid are added in the related work.
And I added a study on the successful attack of MTCNN in the digital domain to the introduction to prove that MTCNN is naturally resistant.
11
It has been corrected.
12
a) b) d) e) f) g)
It has been revised in the text. Thank you for your careful guidance
c)
For example, using deepfakes technology to change the face of politicians and make inappropriate speeches is enough to cause a crisis.
h)
And, after face detection on the image, the user can manually delete some misrecognized faces.
i)
The black line structure fails three face detectors because the continuity of feature extraction is blocked. The three face detectors all extract image features with a CNN structure. Therefore, we have a separate description of CNN.
Thanks again for liking this research.
Round 2
Reviewer 1 Report
Since the authors acknowledge the significant limitations of the paper, it can be accepted.
The approach/results are not currently useful in practice, however they might lead to some interesting approaches.
Reviewer 3 Report
I am satisfied with the authors feedback
Reviewer 4 Report
Thank you for addressing my comments.